Claw shape variation in oribatid mites of the genera Carabodes and Caleremaeus: exploring the interplay of habitat, ecology and phylogenetics

Kerschbaumer Michaela michaela.kerschbaumer@uni-graz.at
Schäffer Sylvia
Pfingstl Tobias
Institute of Biology, University of Graz , Graz , Austria
Gorb Stanislav
Electronic publication date: 2023 Sep 25
Publication date: 2023
Volume: 11
Electronic Location ID: e16021
Received 2023 May 17; Accepted 2023 Aug 11
Copyright: ©2023 Kerschbaumer et al.
Copyright year: 2023
Copyright holder: Kerschbaumer et al.
License: This is an open access article distributed under the terms of the Creative Commons Attribution License, which permits unrestricted use, distribution, reproduction and adaptation in any medium and for any purpose provided that it is properly attributed. For attribution, the original author(s), title, publication source (PeerJ) and either DOI or URL of the article must be cited.
License URL: https://creativecommons.org/licenses/by/4.0/

Keywords: Terrestrial, Geometric morphometrics, Phylogenetic signal, Barcodes, Euryoecious lifstyle

Funding: Austrian Science Fund (FWF) P33869 This study was supported by the Austrian Science Fund (FWF) under Grant [P33869]. The funders had no role in study design, data collection and analysis, decision to publish, or preparation of the manuscript.

==============================
Background

Claws are a commonly observed biological adaptation across a wide range of animal groups. They serve different functions and their link to evolution is challenging to analyze. While there are many studies on the comparative anatomy and morphology of claws in reptiles, birds and several arthropods, knowledge about claws of soil-living oribatid mites, is still limited. Recent research on intertidal oribatid mites has shown that claw shape is strongly correlated with microhabitat and is subject to ecological selective pressures. However, the selective constraints shaping claws in terrestrial oribatid mites are still unknown.

Methods

In this study, 300 specimens from 12 different species and two genera were examined. Geometric morphometrics were used to quantify claw length and curvature, and to analyze two-dimensional claw shape. In combination with molecular phylogenetic analyses of investigated populations phylogenetic signal was quantified within genera using Blomberg’s K and random replicates. Additionally, ecological information on the investigated species was gathered from previous studies and compiled into tables.

Results

The claw shapes of Carabodes species vary moderately, with the three species C. reticulatus, C. rugosior and C. tenuis deviating the most from the others. These three species are only found in a small number of habitats, which may require a more specialized claw shape. Our results show that there is a phylogenetic influence on claw shape in Carabodes but not in Caleremaeus. Additionally, habitat specificity and lifestyle were found to have ecological impact on claw shape in both genera. The present results demonstrate that characteristics of the claws of terrestrial oribatid mites are correlated with ecology, but this correlation is apparently weaker than in intertidal oribatid mites that are prone to strong external forces.

Introduction

Claws are prevalent biological adaptations found in a diverse range of animal groups, including arthropods, birds, reptiles, and large mammals. Those structures can serve various functions (Tinius & Patrick Russell, 2017). The link between claw morphology, its function and evolution are difficult to quantify, analyze, and interpret. Because claws are the most common grip mechanism in vertebrates (Zani, 2000), there are many studies about the comparative anatomy and morphology of these structures, mainly in reptiles (Zani, 2000; Tulli et al., 2009; D’Amore et al., 2019; Alibardi, 2020; Mann et al., 2021; Tulli, Manzano & Abdala, 2022) and birds (Feduccia, 1993; Hahn et al., 2014).

In view of their huge diversity, studies on arthropod claws used for attachment are not as numerous as in vertebrates, but the existing amount of literature is substantial and demonstrates important facts about these morphological structures. The attachment ability of small arthropods is determined by surface roughness and claw tip diameter (e.g., Heethoff & Koerner, 2007; Pattrick, Labonte & Federle, 2018), i.e., sharper claw tips will have more asperities to interlock with, but with heavier loads they are more likely to break. That means larger animals will have blunter claws and thus poorer attachment abilities than smaller animals (Pattrick, Labonte & Federle, 2018). However, insects possess not only claws but also adhesive pads and sometimes other additional structures to maintain proper attachment on different surfaces (e.g., Friedemann, Schneeberg & Beutel, 2014). Claws are basically used on rougher surfaces while pads allow to cling to smooth surfaces (e.g., Gorb, 2001). The complex interplay of these structures allows the insects to walk safely on a huge variety of different surfaces. Other insects are specialized to adhere to specific substrates and in these cases, attachment devices may show remarkable adaptations. For example, parasitic dipteran flies need to stay on the body of the host and therefore have evolved dentate or comb-like claws as well as dichotomously shaped setae on their pads allowing extreme strong attachment (Petersen et al., 2018; Büscher et al., 2022).

There are numerous other studies on the diversity and function of arthropod claws but when it comes to microarthropods like mites, research is relatively scarce. Mites are known to exhibit the largest number of claw characteristics among Chelicerata (Dunlop, 2002) and it was shown that a tiny soil-living moss mite species produces exceptionally high relative claw forces presumably trumping all other organisms (Heethoff & Koerner, 2007). Mites also often show additional adhesive pads for attachment (e.g., Dunlop, 2002) but the majority of soil-dwelling oribatid mites lack these structures (Pfingstl, 2023). There are presently ca. 11,000 known oribatid mite species (Subías, 2022) occupying a wide range of ecological niches, which means the landscape structure is highly variable for the different taxa (Heethoff & Koerner, 2007). This microhabitat variety and the fact that most Oribatida only possess claws for attachment, makes them ideal candidates to investigate the direct correlation between claw characteristics and used substrate.

Pfingstl, Kerschbaumer & Shimano (2020) investigated the claw shapes of numerous intertidal oribatid mites from various habitats by means of geometric morphometrics and their results demonstrated that claw shape strongly correlates with the microhabitat. Species living on rocky shores have remarkably high and strongly curved claws, whereas species from mangrove habitats have significantly lower and less curved claws. Euryoecious species can dwell in a wide range of habitats and show an intermediate claw type. An additional molecular genetic investigation of intertidal species showed that there is no phylogenetic signal in claw shape, which indicates that ecology has acted as one of the primary selective forces in the diversification of claw shapes in intertidal oribatid mites (Kerschbaumer & Pfingstl, 2021). Juveniles of these arthropods exhibit habitat-specific claws. While claw length grows in direct proportion to increasing body size, claw curvature is almost static during development (Pfingstl & Kerschbaumer, 2022). However, these littoral oribatid mites are monodactyl, which means they only possess a single claw on each tarsus, and they are subject to intense wave action and surf, therefore, a strong evolutionary selection for specific claw shapes is assumed (Pfingstl, Kerschbaumer & Shimano, 2020).

In terrestrial oribatid mites there are, next to monodactyl species, also species with two or three claws on each leg, and nothing is known about the selective constraints shaping these claws. Most oribatid species associated with above-ground habitats in forests are considered to have evolved from lineages associated with the forest-floor soil and litter, and thus may have evolved modifications in their morphology in relation to habitat structure and other modifications in life-history traits (e.g., Behan-Pelletier & Walter, 2000). A recent review article (Pfingstl, 2023) highlighted a huge variety of claw expressions in oribatid mites and demonstrated that almost nothing is known about the interaction of these claws with specific environments. Despite ongoing research, little is currently known about the precise reasons behind the development of specific claw formations or the existence of varying numbers of claws.

In this work, we performed qualitative and quantitative analyses to explore possible links between morphological variation and both ecological factors and phylogenetic constraints that could have driven the evolution of claws of monodactyl oribatid mite species in terrestrial habitats. We examined 12 species from Austria belonging to the two oribatid mite genera Carabodes and Caleremaeus.

Carabodes (Acari, Oribatida Carabodidae), a morphologically characteristic oribatid mite genus was originally proposed by Koch in 1835. The type species for this genus is Carabodes coriaceus (Koch, 1835). Currently, the genus includes four subgenera and 135 species that are distributed worldwide (Subías, 2022). These mites can be found in various habitats, including soil, litter, mosses, lichens, fungi, and on the bark of twigs, branches, and tree trunks. They can also occur on rock surfaces and in rotten wood (Reeves, 1987; Reeves & Behan-Pelletier, 1998). As panphytophages, they are not specialized feeders, which accounts for their adaptability to such a wide range of habitats (Reeves, 1987). In Weigmann et al.’s (2015) work on the distribution and ecology of oribatid mites in Germany, one can discover the specific habitats and lifestyles of each Carabodes species. Presently, there are 14 species of Carabodes known to occur in Austria (Krisper, Schatz & Schuster, 2017).

The genus Caleremaeus has recently been reexamined by Norton & Behan-Pelletier (2020), who listed four valid species: Caleremaeus monilipes (Michael, 1882) from the Palaearctic region, as well as C. retractus (Banks, 1947), C. arboricolus (Norton & Behan-Pelletier, 2020) and C. nasutus (Norton & Behan-Pelletier, 2020) all found in North America. The palearctic species Caleremaeus monilipes is highly adaptable and can live in a wide range of environments. It has been observed in various habitats across Europe, including alluvial forests, alpine meadows, spruce forests, deciduous forests, dry grasslands, and scree slopes (Ayyildiz et al., 2011; Schatz, 1983). The species is known to colonize a diverse array of substrates, such as soil, litter, mosses, lichens, decaying wood, and algae. In addition to its ability to live in different habitats and substrates, C. monilipes has also demonstrated a remarkable vertical distribution, ranging from colline to alpine regions. In Austria, it has been recorded at elevations exceeding 2600 m above sea level (Schatz, 1979). In 2021, however, Lienhard & Krisper found out that C. monilipes in central and southern Europe indeed comprises six different species, with five species new to science: Caleremaeus mentobellus, C. lignophilus, C. alpinus, C. elevatus, and C. hispanicus, and all these species differ by their ecological preferences and needs.

We chose members of these two oribatid mite genera for our study because they all show a single tarsal claw, have similar lifestyles, and species of the two different genera could sometimes even be found in the same sample. Both genera might be classified as euryoecious, but certain individual species within these genera inhabit different microhabitats. We want to investigate which claw shapes exist in all these species and find out if they differ between the taxa. If differences are present, are these correlated with diverging ecologies or are they results of phylogenetic relatedness. Basically, this study should give us first insights into the interplay of claw shapes and environment in purely terrestrial oribatid mite species. What shapes claws of terrestrial species and what is the role of ecology?

Material and Methods

Sample and data collection

We examined 300 specimens from 12 different species throughout the investigation (Table 1). We collected data on the genus Caleremaeus (Eremaoidea), including samples from three distinct species: Caleremaeus alpinus, Caleremaeus mentobellus, and Caleremaeus lignophilus; originating from nine different populations. We also gathered data on mites from the genus Carabodes (Carabodoidea), which comprised samples from 18 populations belonging to eight different species, namely Carabodes areolatus, Carabodes coriaceus, Carabodes labyrinthicus, Carabodes marginatus, Carabodes ornatus, Carabodes reticulatus, Carabodes rugosior and Carabodes tenuis. We included two populations of the species Odontocepheus elongatus (Carabodoidea) as a closely related outgroup for morphological investigation of the claw. They were not integrated in the phylogenetic signal analyses.

Table 1 Sample information.

Caleremaeus	Pop ID	Location		Habitat	Date	n	Leg.	
Caleremaeus alpinus	CIA22	Festenburg	Seebach, forest, 1150 m	moss on rock	Oct.20	6	Kerschbaumer	
Caleremaeus alpinus	CIA40	Weinebene	1800m	moss on rock	Aug.20	8	Bodner	
Caleremaeus alpinus	CIA76	Moschkogel	Weinebene, 1750 m	alpin meadow	Jul.22	10	Fröhlich	
Caleremaeus lignophilus	CIA58	Festenburg	Dorfstatt, forest ,900 m	deadwood	Sep.21	10	Kerschbaumer	
Caleremaeus lignophilus	CIA69	Weizklamm	Jägersteig, 600 m	deadwood	May.22	9	Kerschbaumer/Pfingstl	
Caleremaeus lignophilus	CIA74	Weizklamm	Jägersteig, 600 m	deadwood	May.22	10	Kerschbaumer/Pfingstl	
Caleremaeus mentobellus	CIA71	Weizklamm	Jägersteig, 600 m	moss on rock	May.22	9	Kerschbaumer/Pfingstl	
Caleremaeus mentobellus	CIA72	Weizklamm	Jägersteig, 600 m	moss on rock	May.22	7	Kerschbaumer/Pfingstl	
Caleremaeus mentobellus	CIA73	Weizklamm	Jägersteig, 600 m	moss on rock	May.22	6	Kerschbaumer/Pfingstl	
Carabodes	Pop ID	Location		Habitat	Date	n	Leg.	
Carabodes areolatus	CIA03	Graz	Lechwald, forest	deadwood	Dec.20	18	Kerschbaumer/Pfingstl	
Carabodes areolatus	CIA55	Festenburg	Dorfstatt, forest	deadwood	Jun.21	9	Kerschbaumer	
Carabodes areolatus	CIA56	Festenburg	Dorfstatt, forest	deadwood	Jun.21	9	Kerschbaumer	
Carabodes areolatus	CIA74	Weizklamm	Jägersteig, 600 m	deadwood	May.22	10	Kerschbaumer/Pfingstl	
Carabodes coriaceus	CIA09	Vorauer Schwaig	alp, 1500 m	lichen on tree	Aug.20	15	Kerschbaumer	
Carabodes labyrinthicus	CIA09	Vorauer Schwaig	alp, 1500 m	lichen on tree	Aug.20	10	Kerschbaumer	
Carabodes labyrinthicus	CIA21	Vorauer Schwaig	alp, 1500 m	lichen on tree	Oct.20	9	Kerschbaumer	
Carabodes labyrinthicus	CIA45	Festenburg	Seebach, 1000 m	lichen on tree	Apr.21	10	Kerschbaumer	
Carabodes labyrinthicus	CIA47	Festenburg	Seebach, 1000 m	lichen on tree	Apr.21	11	Kerschbaumer	
Carabodes marginatus	CIA12	Vorauer Schwaig	alp, 1500 m	moss, alpin	Aug.20	16	Kerschbaumer	
Carabodes ornatus	CIA42	Weizklamm	Vogelhütte	litter	Mär.21	20	Bodner	
Carabodes ornatus	CIA74	Weizklamm	Jägersteig, 600 m	deadwood	May.22	9	Kerschbaumer/Pfingstl	
Carabodes reticulatus	BF003	Puch/Paldau	Paldau	bracket fungi	Sep.22	7	Schäffer	
Carabodes rugosior	CIA67	Festenburg	Dorfstatt, forest	bracket fungi	May.22	15	Kerschbaumer	
Carabodes rugosior	CIA77	Festenburg	Dorfstatt, forest	bracket fungi	Aug.22	11	Kerschbaumer	
Carabodes tenuis	CIA58	Festenburg	Dreibach,forest	deadwood	Jun.21	15	Kerschbaumer	
Carabodes tenuis	CIA59	Festenburg	Greith, forest	deadwood	Sep.21	4	Kerschbaumer	
Carabodes tenuis	CIA74	Weizklamm	Jägersteig, 600 m	deadwood	May.22	11	Kerschbaumer/Pfingstl	
Odontocepheus elongatus	CIA69	Weizklamm	Jägersteig, 600 m	deadwood	May.22	8	Kerschbaumer/Pfingstl	
Odontocepheus elongatus	CIA74	Weizklamm	Jägersteig, 600 m	deadwood	May.22	8	Kerschbaumer/Pfingstl	

Geometric morphometrics

To perform claw morphometrics, we embedded each specimen in a microscopic slide using lactic acid and then photographed them in dorsal view with a digital microscope (Keyence VHX-5000). Subsequently, we applied pressure to crush the specimen so that the remaining legs with the claws were caught in a lateral position between the object carrier and object slide. To standardize the process, we only photographed and analyzed the claw of the first leg. Using VHX-5000_900F Datenkommunikationssoftware Version 1.6.0.0, we measured the body length and claw length from these photographs (Fig. S4). We recorded the x,y coordinates of three landmarks (LM) and 32 semilandmarks using TpsDig2 (Vers.2.31, Rohlf, 2017). We placed 16 semilandmarks equidistantly along the claw edges dorsally between landmarks 2 and 3, and ventrally between landmarks 1 and 3. Therefore we used the TPSdig function “resample curve—by length”. The new 16 points were computed by linear interpolation along the curve. We provide a scheme for the positioning of landmarks in Pfingstl, Kerschbaumer & Shimano (2020). To enhance the analysis, we eliminated four semilandmarks that reflected positions like LM 1-3, resulting in three landmarks and 28 semilandmarks. The claw curvature was calculated from raw landmark coordinates as the angle between the three landmarks on the inner curvature of the claw (gamma). We analyzed two-dimensional claw shape in R with the package ‘geomorph’ (Baken et al., 2021). We did generalized procrustes analysis (GPA) on our landmarks and semi-landmarks (using function gpagen) and performed principal component analyses (PCA) of shape variation on aligned shapes(gm.prcomp). We tested for differences in shape disparity between populations across all species and both genera using the function morphol.disparity (in ‘geomorph’).

Phylogenetic signal

Extraction of total genomic DNA from single individuals followed the Chelex method given in Schäffer et al. (2018). Standardized protocols were applied for PCR amplification, purification and sequencing (Schäffer et al., 2010; Schäffer et al., 2018). We sequenced the standard COI barcoding region (658 bp) for one specimen of each studied population and verified all sequences by comparisons with known ones from GenBank. We used the standard barcoding primers C_LepFolF and C_LepFolR (Hernández-Triana et al., 2014). The two final datasets included eighteen individuals for Carabodes respectively, nine for Caleremaeus. Sequences were aligned by eye in MEGA v6. (Tamura et al., 2013). Maximum likelihood phylogenies were obtained using IQ-TREE (Nguyen et al., 2015) on the platform PhyloSuite v.1.2.2 (Zhang et al., 2020) under Edge-linked partition model for 5000 ultrafast bootstraps (Minh, Nguyen & Von Haeseler, 2013). PartitionFinder2 (Lanfear et al., 2016) was used to select the best partitioning scheme and evolutionary models for three pre-defined partitions (partitioning by codons) under greedy algorithm. All calculated trees are unrooted. All alignments are available in the Supplemental Material. All sequences used in these reconstructions are available from GenBank under the accession numbers OQ970666 to OQ970692.

Based on the phylogenies generated by IQ-TREE we quantified the phylogenetic signal of claw shape within the two mite genera using Blomberg’s K (Blomberg, Garland Jr & Ives, 2003) with 9,999 random replicates using the R package geomorph (Adams & Otárola-Castillo, 2013) and the physignal function.

Our raw data and R code files are available in the Supplemental Material.

Ecological information

To get insights into the ecology of the genus Carabodes, we created a table (Table 2A) using data of Weigmann et al. (2015), where we list the number of habitats and possible living styles for each investigated Carabodes species. We did the same for the genus Caleremaeus based on data from Lienhard & Krisper (2021) (Table 2B).

Table 2 Habitat specificity.

(A) Carabodes species (B) Caleremaeus species and their habitats and lifestyles in literature.

(A) Carabodes	Habitats (Weigmann et al., 2015)	n	Lifestyle	n	
	AR	AW	AZ	E	GM	GQ	HH	LF	LR	LS	LT	MD	MH	SD	SG	UG	US	WF	WL	WM	WN	WT	WZ		ar	bo	el	ep	li	
Carabodes areolatus						x		x			x		x						x		x			7	x	x	x			3	
Carabodes coriaceus								x	x			x	x					x	x		x	x		9	x	x	x	x		4	
Carabodes labyrinthicus	x	x	x	x	x	x	x	x	x	x	x	x	x			x	x	x	x	x	x	x	x	22	x	x	x	x	x	5	
Carabodes marginatus	x		x	x				x	x		x	x	x	x				x	x	x	x	x	x	16	x	x	x			3	
Carabodes ornatus	x							x	x		x		x		x				x		x	x		10	x	x		x		3	
Carabodes rugosior								x											x		x			4	x	x	x			3	
Carabodes tenuis																					x			2	x	x				2	
Carabodes reticulatus		x																	x		x			3		x				1	
Odontocepheus elongatus		x						x	x										x		x	x		7	x	x				2	
(B) Caleremaeus	Habitats (sample information in Lienhard & Krisper, 2021)	n	Lifestyle	n	
	Alpine meadows and mats	Deadwood	Grass with soil	Lichen	Moss on stone/rock	Moss	Soil and litter		ar	bo	el	ep	li		
Caleremaeus alpinus	x	x	x	x	x	x	x	7	x	x	x			3	
Caleremaeus lignophilus		x						1	x					1	
Caleremaeus mentobellus				x	x	x	x	4		x	x			2	
Notes.

AR alpine meadows and mats

AW subalpine forests

AZ dwarf-shrubs heathlands and tall forb stands in alpine zones

E eurytopic (relevant occurrence in more than 3 habitat types (as S [seashore habitats]

L habitats of open non-forest landscape]

W forests and related habitats],...)

GM bogs and swamp waters

GQ springs, spring runoff

HH caves

LF moist and wet grassland

LR inland salt marshes

LS reed beds, marshes

LT dry grassland and scrub

MD degraded bogs and mires

MH raised bogs, transition mires

SD coastal dunes

SG salt meadows, brackish reeds

UG constructions: buildings, walls

US traffic areas: Railway tracks, roads, pavements, urban squares

WF swamps forests, floodplain forests

WL deciduous(mixed) forests on fresh soils (deciduous trees >50%)

WM bog forests

WN coniferous(mixed) forests (coniferous trees <50%)

WT deciduous (mixed) forests on dry soils (deciduous trees >50%)

WZ dwarf-shrub heathland

ar arboricolous-bark dweller

bo soil dweller

el epilithic—on rocks, stones or walls

ep epiphytic—on plants

li limnic—in freshwater

Results

The body size of Carabodes species ranges from approximately 400–800 µm (Fig. 1A). Populations within each species vary slightly in size but are not significantly different. Claw size correlates well with body size in most species. The ratio of claw length to body length (cl/bl) is conspicuously higher only for C. coriaceus, and C. reticulatus. A regression plot of the two sizes (Fig. S2) shows that there is an apparent jump in claw size at a certain body size. Regarding the angle gamma, measurements range from 75 to 105 degrees. It can be observed that C. reticulatus and C. marginatus have the most widely open claws. On average, C. rugosior has the most curved claw with a curvature angle of around 85°. The results of the principal component analysis (PCA) conducted on the Carabodes dataset indicate that the first two principal components (PC1 = 30.6% and PC2 = 22.8%) account for approximately 52% of the total variation. While there is a considerable overlap between individuals of different species, species means are positioned differently in morphospace, with most species clustering around the intersection of PC1 and PC2 (Fig. 1A). Along PC1, there is only minimal separation. Only C. rugosior and C. areolatus are slightly further in the negative range of PC1 and can thus be distinguished from, for example, C. coriaceus. Notably, the meanshape of the outgroup species O. elongatus is located at the highest positive position on PC2, indicating a claw that is less elongated compared to the seven Carabodes species. The ordination of the specimens along the first two principal components shows that variation along PC2 is mainly related to species affiliation. The corresponding shape changes in the positive or negative direction of PC axis 2 show us a more compact and hunchbacked form of the species that are positioned in the positive range of PC axis two, and a slightly more elongated, drawn-out claw of the species that appear in the negative range.

Figure 1 Clawshape in Carabodes.

(A) Body length, claw length in relation to body length and claw curvature (gamma) for the different species (given in different colors, see legend 1b) and populations (different bars) of Carabodes and the outgroup Odontocepheus elongatus. (B) Scatterplot of principal component analysis showing the first two components (PC1 = 30.6% and PC2 = 22.8%). On the left-hand side, a representative photograph of C. rugosior, one of the Carabodes species, is presented, while at the right-hand side of the scatterplot shape changes associated with PC2 are shown.

The body size of Caleremaeus species ranges from approximately 315–400 µm, with the smallest species examined being C. lignophilus (340 µm), and the largest being C. alpinus (380 µm) and C. mentobellus (360 µm) (Fig. 2A). While populations within each species vary slightly in body size, they are not significantly different. Notably, the ratio of claw length to body length (cl/bl) is lower in C. alpinus. In terms of the angle gamma, measurements range from 80 to 100 degrees, with no marked differences observed among the three species, only C. alpinus shows a slightly lower gamma in all three populations. Results of the principal component analysis(PCA) conducted on the Caleremaeus dataset indicate that the first two principal components (PC1 = 32.34%, PC2 = 21.38%) account for approximately 54% of the total variation. While there is considerable overlap between individuals of different species, species means are positioned differently in morphospace. C. lignophilus and C. mentobellus cluster closely together, while C. alpinus exhibits a different claw mean shape (Fig. 2B). The corresponding shape changes in the positive or negative direction of PC axis 2 indicate a more curved form of the species that are positioned in the positive range, and a slightly more elongated claw of the species that appear in the negative range. There are no marked differences in claw disparity among species in both genera (Fig. S1).

Figure 2 Clawshape in Caleremaeus.

(A) Body length, claw length in relation to body length and claw curvature (gamma) in different species and populations of Caleremaeus. (B) Scatterplot of principal component analysis showing the first two components (PC1 = 32.34%, PC2 = 21.38%). On the left-hand side, a representative photograph of C. lignophilus, one of the Caleremaeus species, is presented, while at the right-hand side of the scatterplot shape changes associated with PC2 are shown.

Phylogenetic influence

By examining the scatterplot in Fig. 3, which shows the average shapes of different populations of Carabodes, we can see that there are distinct groupings. When compared to the phylogenetic tree of the same populations, similarities can be seen. The populations of the species C. marginatus, C. reticulatus, C. coriaceus, and C. ornatus cluster together, while the remaining species form distinct groups. Furthermore, running the Kmult method in R confirms presence of a phylogenetic signal in claw shape for Carabodes (K = 2 × e−5, P = 0.0036). Despite this phylogenetic influence in claw shape, it is possible to observe that there are species standing out in terms of their claw shape (Figs. 1A and 3). C. rugosior, C. reticulatus and C. tenuis exhibit claw shapes that deviate from the mean.

Figure 3 Clawshape and phylogeny in Carabodes.

(A) PCA with mean shapes of all investigated Carabodes species and populations; (B) phylogenetic tree based on COI sequences of investigated Carabodes populations (colors refer to phylogenetic clades).

For Caleremaeus we get another picture, regarding phylogenetic influence. The phylogenetic tree of Caleremaeus populations demonstrates that we have distinct species, with populations of each clustering together (Fig. 4). But in terms of their claw shapes, populations of the same species are not more similar to each other than they are to populations of other species, indicating the absence of a phylogenetic signal. Confirming these findings by using Blomberg’s K, we find no phylogenetic signal in claw shape in the genus Caleremaeus (K = 6 × e−5, P = 0.666).

Figure 4 Clawshape and phylogeny in Caleremaeus.

(A) PCA with mean shapes of all investigated Caleremaeus species and populations; (B) phylogenetic tree based on COI sequences of investigated Caleremaeus populations (colors refer to phylogenetic clades).

Habitat specificity/ecological impact on claw shape

For our eight investigated Carabodes species we found 23 different habitat types and five types of lifestyles in the above-mentioned literature (Table 2). All species could be assigned to the lifestyle of “soil dwellers”. All but one could be denoted as arboricolous, as bark dwellers. While Carabodes labyrinthicus was found in a high number of 22 different habitats, C. tenuis, C. reticulatus and C. rugosior, with two to four different environments seem to be more specific in their choice of habitat. The investigated Caleremaeus species could be assigned to three different lifestyles, namely arboricolous - as bark dweller, soil dweller and epilithic—on rocks, stones and walls. C. alpinus was found in seven different habitats while C. lignophilus was exclusively found in deadwood.

When we examine all species of both genera in a principal component analysis (PCA), we see that the claw shapes of individual species and genera are not very different. (Fig. 5). Although species of Caleremaeus are located more negatively along both PC1 and PC2, they are still within the morphospace of Carabodes. Based on the claw shapes of the respective species, we can see that they are very similar. Only the species C. rugosior, C. reticulatus and C. tenuis exhibit somehow exceptional forms. C. reticulatus exhibits a long and less curved claw. In contrast, C. tenuis possesses a more curved claw, similar to C. rugosior. However, C. rugosior stands out with not only a stronger curvature but also a higher-arched claw. When considering habitat specificity (Table 2) a correlation between “specialized claw” and ”stronger habitat specificity” can be identified for C. rugosior, C. reticulatus and C. tenuis.

Figure 5 Principal component analysis and meanshapes of claws.

Cumulative PCA on mean shapes of all investigated species from both genera Carabodes and Caleremaeus. Numbers within the circles representing the individual species, indicate the number of populated habitats from Table 2.

Discussion

The two mite genera, Caleremaeus and Carabodes, can both be classified as euryoecious, but within the genus, there is some habitat specificity. In terms of habitus, all investigated Carabodes species and the examined closely related Odontocepheus elongatus are easily distinguishable based on their overall morphology. The claw shape varies moderately, with Odontocepheus elongatus standing out due to its much more curved and compact claw than Carabodes species. The claw shapes of the three Carabodes species C. reticulatus, C. rugosior and C. tenuis deviate the most from the others, and interestingly these three species are only found in a small number of habitats (see Table 2). This higher habitat specificity might require a more specialized claw shape but how this “specificity” looks like is not easy to define. After Weigmann et al. (2015), C. tenuis is restricted to bark and soil in coniferous (mixed) forests and our study samples are exclusively from deadwood taken in coniferous (mixed) forests. It is possible that this narrow niche has resulted in the strongly deviating claw shape of this species, their claws are relatively large for the small body size and show a moderate curvature. We observed a similar phenomenon for Lamellovertex caelatus in a former study (Kerschbaumer & Pfingstl, 2023), where the claw shape of this saxicolous species living only in dry mosses and lichens is significantly less curved than in more euryoecious species. In birds, lesser curved claws are significant for ground dwelling species (Birn-Jeffery et al., 2012) and in lizards, species occurring on sandy soils usually show lower curved claws (Tulli et al., 2009). Greater degrees of curvatures, on the other hand, are supposed to be characteristic for tree climbing species, at least in birds and reptiles (e.g., Feduccia, 1993; Tulli et al., 2009). It is not yet known if a similar correlation applies to the Carabodes species. Carabodes reticulatus and C. rugosior were sampled on bracket fungi in this study. Havar, Amundsen & Okland (2014) found C. reticulatus only in dead sporocarps in a spruce forest. They suggest that Carabodes species found in fruiting bodies of wood-decay fungi are primarily living in decomposing wood where fungal food is limited, but they use the opportunity to multiply efficiently in energy-rich sporocarps if these are available. On the other hand, C. rugosior was never found in sporocarps within the study of Havar, Amundsen & Okland (2014) and is described as an inhabitant of soil near tree bases (Sellnick & Forsslund, 1953; Weigmann, 2006). Their incongruent claw shapes, with C. rugosior having a more curved claw (gamma difference ∼10°) than C. reticulatus, may reflect this microhabitat difference. Interestingly, Weigmann et al. (2015) did not describe any Carabodes species as fungicol. The other Carabodes species which can be found in a noticeably wide range of different habitats (see Table 2), like C. labyrinthicus, show claw shapes that are placed more in the center of the morphospace, i.e., they are neither strongly curved, nor extremely weakly curved, they are intermediate so to speak. The same phenomenon was observed in intertidal oribatid mites, where species with wider ecological ranges show also intermediate claw shapes (Pfingstl, Kerschbaumer & Shimano, 2020). Apparently, extreme claw shapes are not selected for if species should be able to dwell in a wider array of microhabitats. Apart from these results, we could also observe that the claws of the first leg of Carabodes species are basically distinctly smaller than the claws of the remaining legs. The reason for this apparent size difference is unknown and needs further detailed investigation.

However, it seems that phylogenetic relationship between species has a stronger influence on claw shape in Carabodes than the habitat. Species of the same genetic cluster are more similar in terms of their claw shape. We could quantify a phylogenetic signal, indicating that ecology does not play the biggest role in shaping Carabodes claws, as was shown for example, in oribatids of the littoral zone (Kerschbaumer & Pfingstl, 2021).

Our results concerning the genus Caleremaeus support the hypothesis of Lienhard & Krisper (2021), who postulate a strong association of the different Caleremaeus species to specific microhabitats. They found a clear genetic differentiation between species of neighboring microhabitats, but not between distant microhabitats of the same type, thus a high degree of habitat specialization is assumed. In respect of claw shapes, no apparent pattern can be found correlating this morphological structure with habitat. Caleremaeus alpinus, which is restricted to subalpine and alpine habitats, differs the most from C. lignophilus and C. mentobellus in terms of claw shape. Their claws are noticeably more curved and smaller in size, which results in a lower ratio of claw length to body length when compared to the other two species, C. lignophilus and C. mentobellus (see Fig. S3). Caleremaeus alpinus can be found in a variety of microhabitats and therefore it is surprising that their claws are not intermediate like the claws of euryoecious Carabodes. C. alpinus is restricted to higher altitudes and thus could be adapted to low temperatures. However, we have no evidence for such a correlation and need much more data in this respect. Caleremaeus lignophilus, on the other hand, is a clear specialist, as it can only be found in deadwood. Their claws are the least curved in comparison to the other species (Carabodes included) and a weaker curvature is supposed to be a feature of claws mainly used on soft substrates, at least in intertidal oribatid mites (Pfingstl, Kerschbaumer & Shimano, 2020) but also in lizard species occurring on sandy soils (Tulli et al., 2009). Dead and rotten wood is clearly a soft substrate and consequently the weaker curvature of C. lignophilus claws may be an adaptation to walking on this underground. Nevertheless, the comparison of only three species with partly overlapping ecologies does not allow us to infer any distinct patterns of correlation between claw shape and habitat. In contrast to Carabodes, mapping the claw shapes onto the molecular phylogeny of populations of these Caleremaeus species results in a lack of a phylogenetic signal. These findings would suggest that the claw morphology of Caleremaeus is likely an adaptation of single species to their unique habitat and lifestyle but given the very low number of investigated species of this genus, the present lack of a phylogenetic signal must be regarded with caution.

Comparing the claw shapes of Caleremaeus and Carabodes, results in surprisingly similar shapes with relatively few divergences between these very distantly related genera (see Fig. 5). Even though single species may show large ecological variances, most of the species occupy similar habitats, and Caleremaeus and Carabodes were often found together in the exact same sample. This indicates that the overlapping habitat preferences result in similar claw shapes in these taxa originating from different superfamilies.

Methodological aspects

In our study, we use 2D instead of 3D morphometrics because capturing the miniature claw in 3D was not feasible with our current resources, given its size of less than 20 µm. However, analyzing claw curvature, claw length, and overall shape from 2D images still provides valuable information for comparing claw shapes among mite species. Although there are limitations to the resolution of the light microscope, we can enhance the photos optically, such as by increasing contrast, to ensure the accurate identification of all crucial points during digitization. 3D morphometrics and higher-resolution imaging techniques would undoubtedly contribute to a more comprehensive understanding of claw shape and curvature in mites. A study conducted by Dai, Gorb & Schwarz (2002) examined the relationship between the dimension of the claw tip and the substrate texture in generating friction force in the beetle Pachnoda marginata. Their findings concluded that the friction force is influenced by both surface roughness and claw tip diameter. Additionally, another study by Pattrick, Labonte & Federle (2018) explored the effect of claw tip diameter on attachment performance in insects on rough surfaces. They proposed that attachment performance decreases with increasing body size due to mechanical constraints on claw design. Applying such investigations to oribatid mites would be highly intriguing. As another methodological aspect, we want to mention that claw curvature varies in its definition across literature, posing challenges for result comparisons. In our study, we adopted a similar approach to Zani (2000), measuring angles between specific points to quantify the inner curvature of the claw. For us this method is straightforward as it utilizes landmark coordinates obtained from shape analysis. For instance, Birn-Jeffery et al. (2012) considered both inner and outer claw curvature angles, while Feduccia (1993) quantified curvature as the central angle formed by radii from claw ends, representing the arc’s portion within the claw. It is important to note that the specific method used to measure claw curvature may vary depending on the study and the organisms being investigated and therefore could not be compared.

Conclusions

The present results demonstrate that characteristics of the claws of terrestrial oribatid mites are correlated with ecology, but this correlation is apparently weaker than in intertidal oribatid mites that are prone to strong external forces. Terrestrial habitats are less exposed to wind and water than coastal environments and the falling of a leaf is not as dramatic for the mites as being washed away into the open ocean. Therefore, selection on claw shape may work to a lesser extent in terrestrial mites. The nature of the correlation of claws with other factors remains largely unclear due to the complex microhabitat features of terrestrial habitats. Further detailed studies on terrestrial species being specialized to certain microhabitats may reveal which claw shapes may be preferable for specific environments.

Supplemental Information

Figure S1 Claw disparity

Claw disparity (procrustes variance) in Carabodes and Caleremaeus species

Click here for additional data file.

Figure S2 CL/BL

Clawsize in relation to bodylength in different Carabodes species.

Click here for additional data file.

Figure S3 Measurements on claw

Measurements in populations and species of both genera.

Click here for additional data file.

Figure S4 Study object and measurements

(A) Dorsal view of Carabodes areolatus with body length measurement and (B) photograph of the first leg and schematic drawing of the claw with landmarks, semilandmarks and measurements.

Click here for additional data file.

Supplemental Information 5 COI sequences for Caleremaeus

Click here for additional data file.

Supplemental Information 6 COI sequences for Carabodes

Click here for additional data file.

Data S1 Geometric morphometrics and phylogenetic analysis

Click here for additional data file.

File S1 Code for landmark analysis and phylogenetic signal in R

Click here for additional data file.

We would like to thank M. Bodner and D. Fröhlich for providing samples for this study.

Additional Information and Declarations

Competing Interests

Author Contributions

DNA Deposition

Data Availability

The authors declare there are no competing interests.

Michaela Kerschbaumer conceived and designed the experiments, performed the experiments, analyzed the data, prepared figures and/or tables, authored or reviewed drafts of the article, and approved the final draft.

Sylvia Schäffer performed the experiments, analyzed the data, authored or reviewed drafts of the article, and approved the final draft.

Tobias Pfingstl conceived and designed the experiments, performed the experiments, authored or reviewed drafts of the article, and approved the final draft.

The following information was supplied regarding the deposition of DNA sequences:

The DNA sequences are available in the Supplemental File. The DNA sequences are provided at GenBank: OQ970666–OQ970692.

The following information was supplied regarding data availability:

The raw data (measurement, landmark coordinates, COI sequences) and code are available in the Supplemental Files.

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
