# Peer review of "Claw shape variation in oribatid mites of the genera Carabodes and Caleremaeus: exploring the interplay of habitat, ecology and phylogenetics"

_PeerJ, doi:10.7717/peerj.16021_

## Round 0.1 · original submission · Major Revisions

Please, especially consider comments of the reviewer 3.

·

Basic reporting

This is a nice, clear and succinct paper, which provides some interesting data and conclusions. Some of the English and punctuation needs a little work (see annotated PDF provided), but these are minor corrections. The ideas and inferences seem sound, and the authors do not appear to have overreached in their conclusions. They appear to take a balanced and cautious approach to their data. The figures and tables are also clear, and the authors have provided plenty of context and background through the literature. The paper is also self-contained and the authors do a good job of connecting their ideas and inferences to the data (with one or two very minor exceptions that are addressed in the annotated PDF).

Experimental design

The experimental design seems solid, with plenty of background and information provided for the experimental design and methods, allowing replication of their work if needed. The research questions were well-defined and the authors are effective in their attempts to address their main questions, which are interesting and worthy of investigation.

Validity of the findings

The results of the paper are interesting and novel, especially because of the contrast the authors highlight with respect to intertidal oribatids. The authors do not over-speculate and so their findings seem valid given their results. The conclusions and inferences are therefore easy to follow, and the authors should be commended on producing a paper that is concise, easy and interesting to read.

Additional comments

No comment.

Reviewer 2 ·

Basic reporting

The lenguaje was clear and professional through all text. The paper includes a solid background to support the main topic of this research. This paper fits in the relative new line of research including geometrical morphometrics in oribatid claws to analize the habitat preferences of some species, which is crucial to understand the communities distribution. It is in a proper scientific format, the figures and tables are relevant Some works cited in the text are not present on references and show relevant results.

Some problems are present, including some woks which are cited on text but are not present in references, (E.g. line 91. Reeves & Behan-Pelletier, 1998; line 96. Krisper et al., 2017; line 103. Ayyildiz et al., 2011)

Experimental design

This article is consistent with the objectives and scope of the journal, the research question is clearly formulated, and the claims are usually well supported. The methods are described in good detail. Review and raw data analysis were confirmed by this reviewer in other software and the fit with the authors was sentenced in the text, with some branches flipping only due to an unrooted tree, but all clades recovered.

I suggest being more descriptive in the methods section, even if an article was cited for that, the previous article doesn't say how the equidistant reference points were placed, if they use some fans or circles and the reference point was situated at the intersection? or by the measurement and subsequent division of the perimeter?

In molecular part, the authos cited Schäffer et al., 2010, 2018), but in these papers they used 2 overlapping fragments od COI, with 2 different set of primers, including one of 704 and other of 674, I think they used the region I for his 658 bp, can be possible to specify the primers used and region?

More Particular comments was attached in the pdf file.

Validity of the findings

All data were provided, they are ribust and with a good statistical analysis, general conclusions are well stated.

More comments on the PDF

Additional comments

The main comments can be seen in the attached PDF, the article covers a relatively new line of research, which is very interesting to correlate the shape of some structures with the phylogenetic signal or other selective pressures. The authors should be more descriptive in some parts (for example, in the methods), check the references and improve some parts that stand out in the pdf.

Annotated reviews are not available for download in order to protect the identity of reviewers who chose to remain anonymous.

·

Basic reporting

This article is well written, and all procedures are clearly described (with a few exceptions highlighted below).

The literature cited and discussed is strongly biased towards work from the authors, which is partially but entirely owed to the fact that not many studies on orbatid mite claws have been completed. However, in their introduction and discussion, it would be both appropriate and beneficial to "widen the net", to give due credit to broadly relevant previous work, and to discuss the findings of this study in the context of previous results in other taxa.

The figures are well prepared, but example images of the "real" claws, either in manuscript figures or in supplementary information are needed.

Experimental design

The study addressed an important and understudied topic; the methods are described clearly, though I strongly encourage the authors to reflect and discuss on the limitations of (i) 2D (instead of 3D) morphometrics, and (ii) explicitly note the spatial resolution of their light microscopy images of what I understand are rather small claws.

The justification for the focus on oribatid mites is somewhat thin – the introduction starts very broad, and then narrows the scope down rapidly, without giving the reader much of a concrete idea as to why oribatid mites are a particularly good choice for approaching the question of interest (the motivation may be as simple as claws are thought to be particularly important, there is a large variation of species in different habitats etc)

L137: Did this procedure result in consistent orientation/alignment of the claw, so that the same plane was photographed across claws which differ in morphology?

L140: Please provide an explicit definition, ideally with schematic as appropriate, for “claw length” and “body length”

L148: Several definitions of claw curvature have been used in the literature – can the authors please provide a brief assessment as to how their definition differs (or does not), and if so, what the justification/advantage of this different definition is?

Validity of the findings

I encourage the authors to place their findings in a context with similar studies in other taxa (lizards, birds, crabs etc). For example, a good number of publications have speculated about the functional significance of claw curvature, and given that this seems to be one of the main variables which varies, it would be helpful and appropriate to discuss these ideas here. Similarly, I suggest the authors at least mention and discuss the functional significance of one important functional parameter they have not measured: the diameter of the claw tip.

Additional comments

L54 – I concur there are few studies on arthropod claws, but there are some (including work on oribatids!) – and these should be at the very least cited, and also discussed as appropriate. A non-exhaustive list of examples:

Dai Z, Gorb S N, Schwarz U. 2002. Roughness-dependent friction force of the tarsal claw system in the beetle Pachnoda marginata (Coleoptera, Scarabaeidae). J Exp Biol, J Exp Biol 205:2479–2488.

Bar-On B. 2023. The effect of structural curvature on the load-bearing characteristics of biomechanical elements. Journal of the Mechanical Behavior of Biomedical Materials, Journal of the Mechanical Behavior of Biomedical Materials 138:105569.

Ditsche-Kuru P, Barthlott W, Koop J H E. 2012. At which surface roughness do claws cling? Investigations with larvae of the running water mayfly Epeorus assimilis(Heptageniidae, Ephemeroptera). Zoology, Zoology 115:379–88.

Wang L X, Zhou Q, Xu S Y. 2010. Role of locust Locusta migratoria manilensis claws and pads in attaching to substrates. Chin Sci Bull, Chin Sci Bull 1–7.

Taylor G, Palmer A, Barton A. 2000. Variation in safety factors of claws within and among six species of Cancer crabs (Decapoda: Brachyura). Biological Journal of the Linnean Society, Biological Journal of the Linnean Society 70:37–62.

Palmer A R, Taylor G M, Barton A. 1999. Cuticle strength and the size-dependence of safety factors in Cancer crab claws. The Biological Bulletin, The Biological Bulletin 196:281–294.

Smith L D, Palmer A R. 1994. Effects of manipulated diet on size and performance of brachyuran crab claws. Science, Science 264:710–712.

Pattrick J G, Labonte D, Federle W. 2018. Scaling of claw sharpness: mechanical constraints reduce attachment performance in larger insects. J Exp Biol, J Exp Biol 221:jeb188391.

Petersen D S, Kreuter N, Heepe L, Büsse S, Wellbrock A H J, Witte K, Gorb S N. 2018. Holding tight to feathers - structural specializations and attachment properties of the avian ectoparasite Crataerina pallida (Diptera, Hippoboscidae). J Exp Biol, J Exp Biol 221:jeb179242. doi:10.1242/jeb.179242

Yang Z X, Liu Z H, Dai Z D. 2014. Surface Texture and Mechanical Behavior of Claw Material in Beetle Dorcus titanus (Coleoptera: Lucanidae). Applied Mechanics and Materials, Applied Mechanics and Materials 461:305–312.

Yamada S B, Boulding E G. 1998. Claw morphology, prey size selection and foraging efficiency in generalist and specialist shell-breaking crabs. Journal of Experimental Marine Biology and Ecology, Journal of Experimental Marine Biology and Ecology 220:191–211.

Büscher T H, Petersen D S, Bijma N N, Bäumler F, Pirk C W W, Büsse S, Heepe L, Gorb S N. 2022. The exceptional attachment ability of the ectoparasitic bee louse Braula coeca (Diptera, Braulidae) on the honeybee. Physiol. Entomol., Physiol. Entomol. 47:83–95. doi:10.1111/phen.12378

Heethoff M, Koerner L. 2007. Small but powerful: the oribatid mite Archegozetes longisetosus Aoki (Acari, Oribatida) produces disproportionately high forces. J Exp Biol, J Exp Biol 210:3036–3042.

Salerno G, Rebora M, Piersanti S, Saitta V, Gorb E, Gorb S. 2023. Coleoptera claws and trichome interlocking. Journal of Comparative Physiology A, Journal of Comparative Physiology A 209:299–312. doi:10.1007/s00359-022-01554-1

In fact, I am not so sure that the general claim that arthropod claws have received less attention holds up to scrutiny – though perhaps the comparative perspective has received less focus.

---

## Round 0.2 · Minor Revisions

Please, consider some minor comments from Reviewer 1.

·

Basic reporting

See my last comment

Experimental design

See my last comment

Validity of the findings

See my last comment

Additional comments

The authors have addressed the problems that were raised by the reviewers. I applaud their cautious approach. They have not over speculated or over interpreted their data. I am slightly concerned about the new section on methodological aspects. This specifically addresses reviewer 3's comment about the importance of 3D over 2D morphometrics. But this new section adds little to the paper, and makes it seem less succinct. As somebody who does a decent amount of 3D morphological work on mites, especially involving confocal, I can vouch for the authors using the better approach of 2D morphometrics. Aside from being expensive and time consuming, 3D techniques like confocal can add various noisy artifacts that you can avoid using brightfield, DIC or phase contrast. Mite claws are also simple enough and easy enough to manipulate into a standard orientation, making a 2D approach seem appropriate for this study. So there is no need to go into extraneous details or excuses on why 3D was not used. This section can be shortened or removed, although I would not much object if it stayed either. Certainly, 3D morphometrics would be well worth doing if the field was saturated with 2D morphometric studies and there were only a small number of species and forms to study. Neither is the case with Oribatida or most other groups of mites.

Reviewer 2 ·

Basic reporting

The article was improved in all the requested sections

The language used is professional, clear and concise, as are the references, even self-citations are necessary due to the low number of works on this subject by other authors

Experimental design

The article was improved in all the requested sections

The scientific question, its approach and resolution make both statistical and biological sense, although the methods could perhaps be improved, they are very detailed and reproducible.

Validity of the findings

The results of the research are as novel as the subject itself, it provides an advance in the understanding of the form and ecological function of the claws of these organisms and their adaptation to different environments.
The conclusions are based on the results obtained and have a biological and statistical support.

Additional comments

The authors have made the suggested modifications, or failing that, have validly and reasonably justified or explained the reasons for their methods, the comment tracking file was quite useful and well organized.

·

Basic reporting

All good.

Experimental design

All good.

Validity of the findings

All good

Additional comments

The authors have adequately addressed my comments. I am looking forward to seeing this published.

---

## Round 0.3 · accepted · Accept

The authors provided all necessary changes in the current version of the manuscript, which can be accepted.